# Aerosol Composition Retrieval from a combination of three different space-borne Instruments: Information content analysis

Ulrike Stöffelmair[1,2], Thomas Popp[1], Marco Vountas[2], and Hartmut Bösch[2]

[1]German Remote Sensing Data Center (DFD), German Aerospace Center (DLR), 82234 Oberpfaffenhofen, Germany
[2]University of Bremen, Institute of Environmental Physics (IUP), Otto-Hahn-Allee 1, 28359 Bremen, Germany

**Correspondence:** Ulrike Stöffelmair (ulrike.stoeffelmair@dlr.de)

**Abstract.** This study focuses on the information content for retrieving Aerosol Optical Depth (AOD) and its components from satellite measurements. We utilize an optimal estimation retrieval algorithm with data from three satellite-based instruments: SLSTR on Sentinel 3A/3B, IASI and GOME-2 on MetOp A/B/C. Data are averaged to a common 40x80 $km^2$ grid, temporally aligned within a 60-minute window and cloud masked. A simulation study has been carried out to analyse the information
content of the instrument combination, identify retrievable parameters and initiate the development of a uniform retrieval algorithm for the AOD and aerosol components. The simulation study for the information content analysis is implemented using the radiative transfer model SCIATRAN and uses MERRA-2 reanalysis data for AOD and mass mixing ratios of different aerosol components. The study shows 6 to 15 degrees of freedom for the determination of aerosol components dependent on AOD and the underlying surface. The results will be used for the development of a synergistic multi-sensor retrieval algorithm
for AOD and its components in cloud-free atmospheres across various surface types.

## 1   Introduction

Aerosols impact radiation and climate in a multitude of ways and, after CO2, their combined effects are the second largest contributor to the radiative forcing. Due to the complexity of the aerosol-climate effects, aerosols are the largest contributor to uncertainties (Forster et al., 2021; Li et al., 2022).

Their direct (influencing the radiation budget directly by scattering, absorbing or emitting radiation), semi-direct (effects on cloud properties by heating or cooling the atmosphere) and indirect effects (affecting cloud properties through acting as a condensation nuclei or ice nucleating particles) depend not only on the aerosol abundance and geospatial distribution but also on the aerosol chemical composition (Boucher et al., 2013; Kaufman et al., 2002). Direct radiative effects will lead to warming for strongly absorbing aerosols e.g. dust, black and organic carbon absorbing aerosols ... (Matsui et al., 2018; Samset

et al., 2018; Kok et al., 2023) whereas for most other aerosols will reduce the radiation and energy input into the atmosphere by reflecting solar radiation and thus have a cooling effect (Charlson et al., 1992; Kaufman et al., 2002; Arias et al., 2021; Li et al., 2022). The indirect effects due to aerosol cloud interactions are due to aerosol particles acting as condensation nuclei for water droplets and nucleating particles for ice crystals in clouds, which also depends on the aerosol composition (Twomey, 1974, 1977; Burrows et al., 2022; Seinfeld et al., 2016; Storelvmo, 2017). Other indirect effects are related to changes in sur-

face albedo due to deposited aerosols. An example of the semidirect effect is the uneven distribution of radiative heating in the troposphere caused by aerosols, which leads to atmospheric convection and circulation (Sherwood et al., 2015). All these effects depend on the detailed characteristics of the atmospheric aerosol (Kaufman et al., 2002; Yin et al., 2002; Wiacek et al., 2010; Arias et al., 2021; Forster et al., 2021; Kok et al., 2023). Atmospheric aerosol is typically described by a mixture of a manageably small number of representative components. A component groups particles with similar characteristics (chemical composition, size range, shape and corresponding optical properties). Further we differ between organic and black carbon, sulfates and sea salt and mineral dust at different size bins (Kinne et al., 2006; Randles et al., 2017). Due to the major effect of aerosols and their composition on the climate, they play a critical role in climate modelling (Myhre et al., 2017; Gliß et al., 2021; Randles et al., 2017) , so observational data are important for validation and assimilation purpose. It is not sufficient to constrain just the quantity and distribution of aerosol, composition information is also needed if we want to reduce uncertainties on climate forcing due to aerosol. Hence, there is an important climate research need for global monitoring of aerosol composition from satellite measurements (Kaufman et al., 2002; Holzer-Popp et al., 2008; Li et al., 2022; Kacenelenbogen et al., 2022).

Typically, the retrieval of AOD from satellite measurements is an ill-posed mathematical inversion problem which limits the capabilities to identify components of the AOD. This means that there is not enough information about aerosols and their composition, other atmospheric parameters such as temperature, pressure or trace gas concentrations and surface properties in the measurement data. By combining data from different satellite instruments, complementary information, like varying spectral ranges and observation geometries, can be combined (Dubovik et al., 2021b), which holds the potential to improve the capabilities for inferring AOD and composition. For this purpose a retrieval algorithm based on optimal estimation (Rodgers, 2000) will be developed which makes use of data from three different instruments measuring with different observation characteristics, different spectral ranges (ultraviolett (UV), visible (VIS) and thermal infrared (TIR)) and different viewing geometries (nadir and oblique).

The instruments included are the dual-view radiometer SLSTR (Sea and Land Surface Temperature Radiometer) onboard Sentinel 3A and 3B (Coppo et al., 2010), the Infrared Atmospheric Sounding Interferometer (IASI) (Blumstein et al., 2004) and the spectrometer Global Ozone Monitoring Experiment-2 (GOME-2) (Munro et al., 2006; Callies et al., 2000), both onboard MetOp A/B/C.
The SLSTR measurement with nine channels in the VIS and TIR provides additional information due to the two different viewing directions (nadir and oblique) and therefor provides options for better separation of ground and atmospheric influences. IASI is mostly sensitive to mineral dust and larger particles. The measurement in the UV range of GOME-2 provides information about the absorption and thus enables the separation of absorbing and non-absorbing particles. In addition IASI and GOME-2 are also sensitive to elevated stratospheric sulphate aerosol loadings.
Due to the partial overlap of the wavelength ranges, the SLSTR measurements can be used to check the spectral consistency of the 3 instruments and thus filter out pixels with changes within the half-hour time offset in the overflight that the two satellites have. The temporal overlap of GOME-2 and IASI (both on MetOp A/B/C) is important for the combination of the instruments

due to the rapid possible changes in aerosol and cloud distribution and in our case outweighs the advantages that Sentinel 5P, for example, has with a similar wavelength range to GOME-2 but with significantly better resolution. The planned retrieval is designed for the coarsest resolution of the instruments used (GOME-2 with $40\text{km} \times 80\text{km}$), on which the other instruments are averaged to.

To analyse the aerosol components for climate research, the longest possible time series is required. The algorithm proposed here has the potential to be applied to predecessor instruments (A)ATSR(2) at SLSTR, GOME and SCIAMACHY at GOME-2 and HIRS at IASI, which provide temporal coverage from 1995 to the present with one interruption (2012 to 2016) (Coppo et al., 2010; Loyola et al., 2009; Inamdar et al., 2023). As all the instruments have planned successors the time series can be continued at least until 2035. The planned successor to GOME-2 is UV/VIS/NIR/SWIR Sounding (COPERNICUS Sentinel-5 UVNS) and to IASI it is the Infrared Atmospheric Sounding Interferometer – New Generation both on MetOp second Generation (Holmlund et al., 2017) and SLSTR will be continued on Sentinel 3C and 3D (World Meteorological Organization (WMO), a, b). The selected resolution is sufficiently fine for the planned use of the retrieval for climate studies, while it would be too coarse for regional air quality analyses.

There exist aerosol retrieval algorithms exploiting data of single instruments, but they are not retrieving aerosol components. For example, the AOD and the Fine mode AOD can be determined from SLSTR data (Sayer et al., 2010; Bevan et al., 2012; Sogacheva et al., 2017), the Absorbing Aerosol Index from GOME-2 (Hasekamp et al., 2004) and Dust AOD from IASI (Vandenbussche et al., 2013; Callewaert et al., 2019; Clarisse et al., 2019; Capelle et al., 2014; Klüser et al., 2012). There are also algorithms using a combination of different instruments types to determined more information about the aerosol composition for example SYNAER (Holzer-Popp and Schroedter, 1999; Holzer-Popp et al., 2008), which uses AATSR (Advanced Along Track Scanning Radiometer) and SCIAMACHY (Scanning Imaging Absorption Spectrometer for Atmospheric Chartography) both on Envisat, predecessor instruments to SLSTR and GOME-2 and PMAp (Grzegorski et al., 2021), which uses the polarization channels of GOME-2, IASI and AVHRR (Advanced Very High Resolution Radiometer).

SYNAER works with predefined aerosol mixtures (fixed mixes of different aerosol components), determines these and not individual aerosol components (Holzer-Popp et al., 2008). PMAp works with aerosol classes like oceanic, industrial, biomass and dust with different refractive indices and different size distributions (Grzegorski et al., 2021). Another aerosol component algorithm is GRASP/Component (Li et al., 2019, 2020; Zhang et al., 2021; Dubovik et al., 2021a), which is based on the multi-axis and polarimetric data from POLDER/PARASOL.

As a first step towards developing this multi-sensor retrieval algorithm a simulation-based information content analysis is presented in this study. An information content study shows the amount and type of information which can be extracted from the data. In this context, the degrees of freedom (DGF) represents the number of parameters that can be retrieved.

For SYNAER the information content for aerosol type determination is shown to be 2 to 3 DGF for fixed AOD and surface

albedo (Martynenko et al., 2010) using a principle component analysis. According to Klüser et al. (2015), depending on the spectral database of optical properties, AOD and dust layer temperature used, values of up to 6.7 prevail for the DGF for determining the dust AOD, dust particle size, composition, emission temperature and height. Hasekamp and Landgraf (2005) show that aerosol retrieval based on simulated polarised and unpolarised GOME-2 measurements over the ocean provides 6 to 8 total degrees of freedom calculated with the optimal estimation method, reducing to 3.5 to 5 when only considering the intensity measurements. The determined parameters were the aerosol loading of both modes of the bimodal aerosol size distribution, the effective radius of at least one mode, the refractive index (real and imaginary part), the aerosol layer height and the oceanic pigment concentration.

Consistent with the planned retrieval setup, an information content analysis is performed in this study with simulated cloud-free pixels with realistic observation geometry for the instrument ensemble. A set of observations resembling the above mentioning instruments is simulated with the SCIATRAN radiative transfer model (Rozanov et al., 2014; Mei et al., 2023) for different observing conditions / geometries, surface types, aerosol compositions and aerosol amounts in realistic scenarios. With these data an analysis of the combined information content is then conducted which focuses on capabilities for the determination of aerosol abundance (total AOD) and aerosol components in a cloud-free atmosphere. This analysis uses the Optimal Estimation Theory developed by Rodgers (2000) to calculate the degrees of freedom. This information content analysis will be used to theoretically identify which parameters can be retrieved from the multi-sensor data. It will then be used to develop a synergistic multi-sensor retrieval algorithm for AOD and aerosol components. The planned synergistic retrieval focuses on the determination of the aerosol composition for further investigation of the described climate influences.

This paper starts with a brief theory of information content (Sec. 2) to specify the used definitions. Simulated satellite measurements are used so that the true values are known and are described in Sec. 3. The used methods for the quantitative analysis of the information content is explained in Sec. 4. Section 5.2 presents the results for the information gain and the information content of the instrument combination, which are finally discussed in Sec. 6.

## 2 Theory of information content and Optimal Estimation

The Optimal Estimation theory (Rodgers, 2000, 1996; Maahn et al., 2020) describes the forward model as

$$\mathbf{y} = \mathbf{F}(\mathbf{x}) + \epsilon, \tag{1}$$

where $\mathbf{y}$ represents the observation vector, i.e. the vector which contains the individual measurements in our case spectral reflectance and brightness temperatures , $\mathbf{x}$ the state vector, which contains the parameters that will be retrieved, $\mathbf{F}$ the forward model, which is in our case the radiative transfer model SCIATRAN and $\epsilon$ is the experimental error including observation noise and forward model uncertainty.

For the information content analysis the averaging kernel matrix

$$\mathbf{A} = \frac{\partial \hat{\mathbf{x}}}{\partial \mathbf{x}} = (\mathbf{K}^T \mathbf{S}_\epsilon^{-1} \mathbf{K} + \mathbf{S}_a^{-1})^{-1} \mathbf{K}^T \mathbf{S}_\epsilon^{-1} \mathbf{K} \tag{2}$$

is used to calculate the Degrees of Freedom (DGF). It represents the partial derivation of the retrieval state vector $\hat{\mathbf{x}}$, which is the estimate of the true state vector $\mathbf{x}$ obtained by the optimal estimation algorithm, with respect to $\mathbf{x}$. $\mathbf{S}_a$ is the error covariance matrix corresponding to a priori state vector $\mathbf{x}_a$. The error covariance matrix for the measurements $\mathbf{S}_\epsilon$ contains the instrument measurement uncertainties. $\mathbf{K}$ is the Jacobian matrix consisting of the partial derivatives of each measurement, in this study each calculated $\mathbf{y}$ value from the forward model, with respect to each state element ($\mathbf{K}_{ij} = \frac{\partial y_j}{\partial x_i}$). The superscripts "T" and "-1" refer to matrix transpose and inversion.

The total DGF is calculated as

$$\text{DGF}_{\text{total}} = \text{Trace}(\mathbf{A}) = \sum_{i=1}^{n} \mathbf{A}_{ii} \qquad (3)$$

and the diagonal elements of $\mathbf{A}$ represent the DGF per element of the state vector $\mathbf{x}$:

$$\text{DGF}_i = \mathbf{A}_{ii} = \frac{\partial \hat{x}_i}{\partial x_i} \qquad (4)$$

The diagonal element values of $\mathbf{A}_{ii}$ are in the range of 0 (no information on $x_i$) to 1 ($x_i$ can be fully determined) and characterize the sensitivity of each retrieved parameter to its truth. This makes the DGF a good indicator of the number of parameters that can be determined in retrieval. The off-diagonal elements describing the cross-correlation between the parameters indicate how strongly the estimate of one parameter depends on other parameters.

## 3  Simulation of Satellite Measurements

For the simulation of satellite measurements the radiative transfer model SCIATRAN (Rozanov, 2022; Rozanov et al., 2014; Mei et al., 2023), is used to simulate the collocated data from the three instruments on common GOME-2 pixels: SLSTR on Sentinel 3-A and 3-B (since 2017), IASI and GOME-2 both on MetOp A/B/C (since 2007).

### 3.1  Radiative transfer forward model

We use SCIATRAN as a forward model to create synthetic measurements for this information content study. SCIATRAN simulates radiance spectra appropriate to atmospheric remote sensing observations across the UV to TIR. SCIATRAN is a well test radiative transfer model that can be used to compute radiances over a broad spectral range, in particular the UV-VIS and IR ranges needed for this study. A second reason is SCIATRAN's option to take into account different aerosol components as defined in MERRA-2. We calculate the radiance at the top of atmosphere (TOA) using the assumption of a pseudo-spherical atmosphere with the scalar discrete ordinate technique. For the solar spectrum the Thekaekara (NREL; Drummond and Thekaekara, 1973) spectrum is used.

The surface is handled as a Lambertian reflector with wavelength dependent albedo. The wavelength-dependent emissivity of the surface is set to the climatology values measured with IASI since 2008 (Zhou et al., 2011, 2013, 2018, 2021). The influence of aerosols on TOA radiation is calculated using the different aerosol components defined in the MERRA-2 dataset. The optical database in the MERRA-2 model comprises precomputed values for extinction, scattering efficiency, and expansion

coefficients of scattering matrix elements at specific wavelengths and humidity levels as well as cross-sectional area and particle mass at predefined humidity levels for 15 aerosol components. These components include hydrophobic and hydrophilic modes of black carbon (BCPHOBIC, BCPHILIC) and organic carbon (OCPHOBIC, OCPHILIC), sulfate (SO4), and five distinct size bins for sea salt (SS001, ..., SS005) and dust (DU001, ..., DU005) aerosols (Randles et al., 2017). The aerosol components are listed with their dry effective radius in Tab. A1.

## 3.2 Satellite Measurements and Observation Vector

Satellite TOA radiance data for the three instruments are averaged at a common grid of $40 \times 80$ km$^2$ within a temporal matching window of 60 minutes for the planned retrieval. The choice of instruments has been made with the goal that they complement each other in terms of their information content since they measure in different spectral ranges and with different viewing geometries. In addition the spatial and temporal overlap of their measurements plays an import role in that choice. Moreover, the chosen combination of instruments allows for the possibility of a long time series through their predecessor and successor instruments.

The DGF analysis is made with a perspective of a synergistic retrieval algorithm for those three instruments using atmospheric radiative transfer simulations, which do not include an instrument model, i.e. they are done monochromatically at central spectral bins and one-dimensional. A short overview over the instruments, their characteristics and their contribution to the observation vector is listed below.

### 3.2.1 Sea and Land Surface Temperature Radiometer (SLSTR)

SLSTR measures with the dual-view principle observing the same spot on Earth twice along track in nadir and oblique (rearward) view (55°) and accordingly with two different path lengths through the atmosphere, which enables a better decoupling of the radiation contributions from the ground and the atmosphere (Barton et al., 1989). SLSTR has a swath width of approximate 750km and a spatial resolution of 0.5kmx0.5km in the VIS and SWIR range and 1kmx1km in the thermal range for nadir pixels (Coppo et al., 2010; European Space Agency, 2024). The different measurement channels, listed in European Space Agency (2024) with their central wavelength and bandwidth, are all used in the observation vector with both viewing directions. For the different channels the radiative transfer calculations are performed on an internal wavelength grid and averaged using a rectangular function as a simplification of the spectral response function. For the instrumental error for the SLSTR radiance channels is 5% of the measurements and for the infrared channels 0.5K (European Space Agency, 2024), those are used in the as diagonal elements in the measurement error covariance matrix $S_y$.

### 3.2.2 Infrared Atmospheric Sounding Interferometer (IASI)

IASI is a passive infrared Fourier Transform Spectrometer (FTS), which measures in a spectral range from 3.7 to 15.5μm using a Michelson interferometer, and an infrared camera connected to it, which works in the spectral range from 10.3 to 12.5μm. Through an inverse Fourier transformation and radiometric calibration, a spectrum is calculated in the IASI instrument. IASI

has a swath width of 2400km. The IASI footprints are in nadir circles with about 12km diameter (Blumstein et al., 2004; Hébert et al., 2017; Simeoni et al., 2004).

The precise measurements in the infrared spectral range contain information about the quantity and properties of the dust aerosol particles.

Following Vandenbussche (2021) the observation vector contains IASI data in the original spectral resolution ($0.25\text{cm}^{-1}$) from 8914.6 to 9111.6nm and from 10796.2 to 11061.9nm. For the used Level 1C data a spectral harmonisation removing the impact of instrument spectral response from the radiance spectra Clerbaux et al. (2009) (EUMETSAT, 2019). For IASI the measurement uncertainties are supposed to be below 0.2K (Clerbaux et al., 2009). Following the approach of Vandenbussche (2021), we use 0.5K as the diagonal elements in the $S_y$ matrix.

### 3.2.3 Global Ozone Monitoring Experiment-2 (GOME-2)

GOME-2 on board of MetOp A/B/C is the improved version of the Global Ozone Monitoring Experiment on the second European Remote Sensing Satellite (GOME on ERS-2). GOME-2 is an optical spectrometer with 4096 channels in four bands in the range of $240 - 790$nm with a high spectral resolution of $0.26\breve{\ }0.51$nm and a spectral range for each pixel form 0.07 to 0.2 (EUMETSAT, 2022). A scan mirror enables across-track scanning in nadir direction with a swath width of 1920km and can also be directed to different calibration sources. It has a footprint of 80km (across-track) x 40km (along-track) (Munro et al., 2016; EUMETSAT AC SAF), which is the driver for the resolution of the study and the planned retrieval.

From GOME-2 the data from 342.33nm to 792.40nm with a wavelength step of approximately 10nm is used in the observation vector. A higher resolution is not necessary because aerosols do not have sharp absorption lines but broad structures (Andersson, 2017). The measurement uncertainty is 2% for GOME-2 (EUMETSAT, 2005), which we use in the $S_y$ matrix.

### 3.3 State Vector

The state vector analysed for its information content contains the different parameters which shall be inverted in the retrieval algorithm - and to be developed in the future. In this study these are 25 parameters containing: surface albedo values at different wavelengths (340nm, 494nm, 555nm, 670nm, 758nm, 868nm, 2500nm), the surface temperature, the AOD at 550nm and the scaling factors for the different 15 aerosol components (Tab. A1). Each provided aerosol component height profile is assumed to be scalable with one parameter, in the following called the scaling factor. The scaling factors are applied to to $1\text{kgkg}^{-1}$ normalized vertical mass mixing ratio profiles, taken from monthly mean MERRA-2 data, to obtain the profile of the aerosol components mass mixing ratio. This means that the sensitivity to aerosol components is a mixture of the composition/optical properties of each component and its typical height profile.

### 3.4 Apriori values and error covariance matrix used in Optimal Estimation

We use the following data as a priori values for the described parameters of the state vector: For the surface albedo values from 340 to 758nm we use the climatological values of the GOME-2 surface LER database (Tilstra et al., 2017, 2021) for the geographical position. For the albedo values at 868nm and 2500nm, we set the a priori values to 0.2 and 0.15, as we do not have any climatological values here. The a priori value of the surface temperature is set to 295K. For the AOD and the scaling factors of the aerosol components, the monthly mean values from MERRA-2 data (Global Modeling And Assimilation Office and Pawson, 2015a, b) are used.

The a priori error covariance matrix $\mathbf{S_a}$ has the following diagonal elements: 0.2 for the constrains for the albedo values, 5.0 for the AOD, 30 (K) for the surface temperature and 1 for the scaling factors for the mass mixing ratios of the aerosol components. We use $\mathbf{S_a}$ as a diagonal matrix. Consequently, all off-diagonal elements are set to zero, because the constrain of one parameter to another one is not known.

### 4 Method for quantitative analysis of the information content

In this study we use the Optimal Estimation theory, described in Sec. 2, and the pyOptimalEstimation package (Maahn et al., 2020) to calculate the information content of the forward model arrangement described above. To work with realistic aerosol composition and AOD values in the simulated scenarios we use MERRA-2 data for AOD (Global Modeling And Assimilation Office and Pawson, 2015a) and mass mixing ratios of the different aerosol components (Global Modeling And Assimilation Office and Pawson, 2015b).

To account for a representative range of the true state parameters, global scenarios derived from the MERRA-2 reanalysis are utilized. As we like to do this analysis on a 1°x1° grid, the mass mixing ratios from MERRA-2 are re-gridded from 0.5°x0.625° to 1°x1° using bilinear interpolation. The satellite overpasses are at 9:30 am for the MetOp satellite, with GOME-2 and IASI on it, and 10:00 am for Sentinel 3A and 3B, with SLSTR on board. Consequently, we select for each time zone, every 3 hours in the MERRA-2 data, the nearest to 9:30 am local solar time. The scaling factors are then calculated by normalizing the profiles to $1\mathrm{kgkg}^{-1}$, as described in Section 3.3. The scaling factors are used for the simulation study to calculate the simulated data. The relative humidity is taken from MERRA-2 and is not retrieved in this study.

We use solar angles calculated with the python package pvlib (Anderson et al., 2023) at a local solar time of 9:30 am which corresponds approximately to the satellite overpasses. The minimal solar movements between the satellite overpasses of MetOp and Sentinel-3, which are at most half an hour apart, are neglected in this context. For the satellite viewing geometry, we use the simplified case that all instruments measure as close as possible near nadir above the point under consideration. That means 0° viewing zenith angle for GOME-2 and IASI and 6° for the nadir view of SLSTR.

## 5    Results and Analysis

In order to consider a representative range of all parameters global scenarios based on the MERRA-2 reanalysis are used.
To analyse the gain in information content (Sec. 5.1) of the instrument combination compared to the individual instruments, the data of the 1st, 15th and 30th of the middle months of each season (January, April, July and October) of the year 2018 are used. For this analysis we select a subset of 1°x1° grid boxes to get a 10°x10° grid in order to consider the best possible coverage of different aerosol compositions with a reduced amount of data. This results in 6948 simulated scenarios for each individual instrument and for the combination.

For the more detailed analysis of the information content of the instrument combination (Sec. 5.2), we use data from the 15th of January, April, July and October on the 1°x1° grid. This larger dataset, 231805 simulated scenarios for this consideration, also allows us to recognise patterns on global maps of degrees of freedom.

### 5.1    Increase in information content through the combination of instruments

To analyse the increase in information content from the combination of instruments compared to the individual instruments, we use 6948 simulated scenarios designed like specified in Sec. 4.

The analysis of the total DGF shows a clear gain in information from the total degrees of freedom of the individual instruments ranging from 5 to 8 for GOME-2, from 8 to 13 for SLSTR and from 4 to 10 for IASI up to 14 to 20 DGF by combining the instruments, as shown in Fig.1.

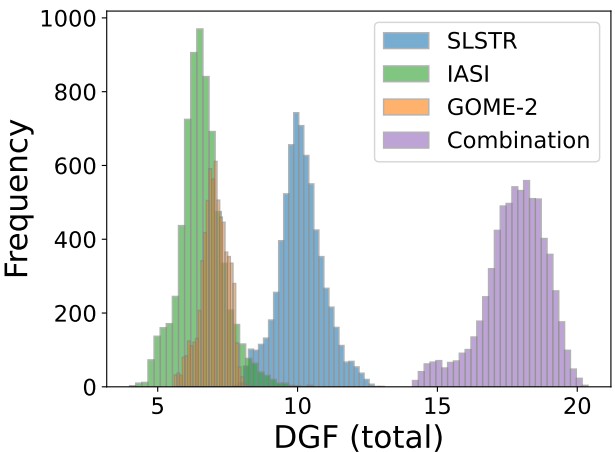

**Figure 1.** Histogram showing the distribution of the total DGF for the individual instruments (SLSTR in blue from 8 to 13, GOME-2 in orange from 5 to 8 and IASI in green from 4 to 10) and for the combination of these instruments in purple from 14 to 20. This demonstrates a significant gain in the total information content achieved through the combination of instruments.

The degrees of freedom for all aerosol components is defined as the sum of the diagonal elements of the averaging kernel matrix corresponding to the MERRA-2 scaling factors for each component ; it provides the result targeted at the primary goal of this study, determining the aerosol composition. A clear gain in information content is also evident here - from 1 to 6 for GOME-2, 3 to 12 for SLSTR and 2 to 10 for IASI to 6 to 15 for the combination of instruments, as can be seen in Fig. 2.

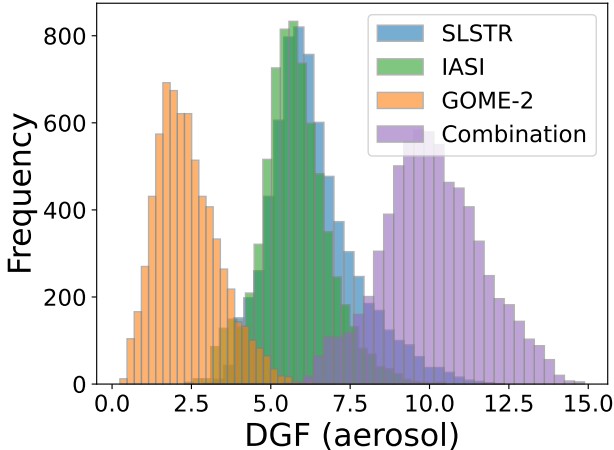

**Figure 2.** Histogram showing the distribution of the sum of DGF for the aerosol components for the individual instruments (SLSTR in blue from 3 to 12, GOME-2 in orange from 1 to 6 and IASI in green from 2 to 10) and for the combination of these instruments in purple from 6 to 15. This shows a significant information gain for the retrieval of aerosol components obtained through the combination of instruments.

This increase in information content through the combination of these three instruments shows that complementary infor-
mation adds up to a more detailed picture of the aerosol properties in the atmosphere. This proves the potential for the planned aerosol retrieval to allow retrieving information on the aerosol composition, i.e. their contributions to total AOD.

### 5.2    Information content analyses for the combined datasets

Analyzing the information content of data gridded at a higher spatial resolution (1°x1°) for only the 15th of each month reveals the following correlations. They show that an increase in the degrees of freedom for aerosol component determination can be
observed following a logarithmic-shape function with increasing AOD, see Fig. 3 center. This means that with a higher aerosol load, more aerosol components can be separated from each other. This is consistent with the results of e.g. Martynenko et al. (2010); Hasekamp and Landgraf (2005); Hou et al. (2017). When considering the degrees of freedom used to determine the surface albedo as a function of AOD, an opposite behaviour is observed - a decreasing number of degrees of freedom for the spectral albedo determination with increasing AOD, as can be seen in Fig. 3 right. The determination of surface albedo becomes
more difficult at higher aerosol loading, because of the shielding effect of aerosols, which is well known and is described for example in Köpke (2012); Popp (1995). For the total degrees of freedom, there is also a logarithmic increase up to approx. AOD = 0.3, which turns into a slow linear decrease at higher AOD values, see Fig. 3 left. This behaviour can be well explained

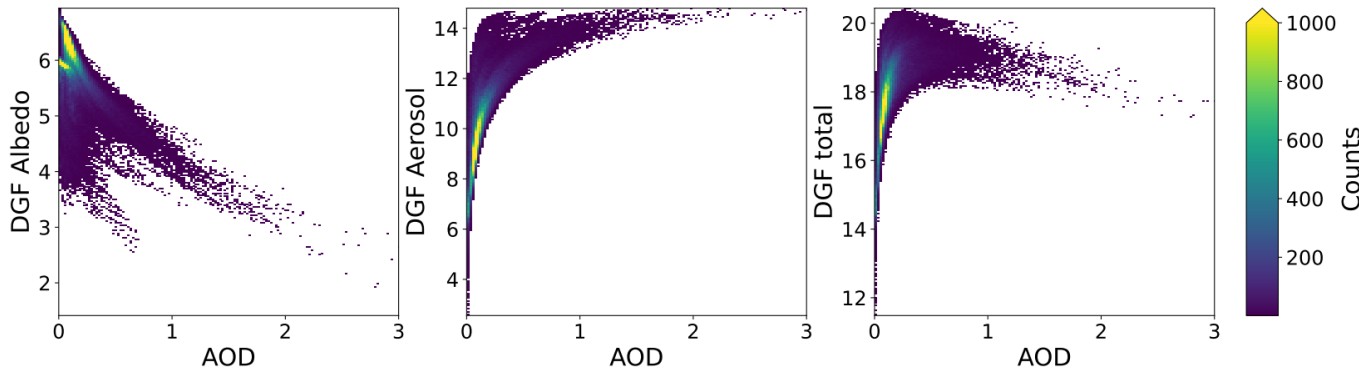

**Figure 3.** 2D histogram showing the relationship between the total DGF (right), the DGF for the aerosol components (center) and the DGF for albedo (left) and the AOD. The degrees of freedom for the determination of albedo decrease as expected with increasing AOD. In contrast, the degrees of freedom for the determination of aerosol components increase with higher AOD. For the total DGF, a rising trend with increasing AOD is observed at low AOD values, which reverses at an AOD of approximately 0.3. This behavior can be explained as the sum of the two partial trends shown in the other subplots.

as a combination of the two contributing pieces for aerosol and surface information.

As an example the maps in Fig. 4 depict the distribution of the total DGF (top left), the DGF of the aerosol components (top right), the DGF of the albedo retrieval (lower left) and the AOD (lower right) at the 15th of July. Data south of 62°S were measured at solar zenith angles of over 90° on this date in north-hemispheric summer. Under these conditions, the amount of reflected UV and VIS radiation is too low and the retrieval is not performed. It is obvious that in areas with high aerosol amounts (such as over the Sahara and over the Atlantic within the Sahara dust plume), more degrees of freedom are identified for the determination of aerosol components. Also, over the Roaring Forties and Furious Fifties between 40° and 60° S higher DGFs for the determination of aerosol components appear which are probably related to the higher concentration of larger sea salt aerosol components near their source regions. In contrast, we see an opposite effect with similar spatial pattern when considering the degrees of freedom for albedo determination.

The observation of higher DGF for aerosol components over areas with large sea salt or dust aerosol fraction at comparatively low total AOD is also supported by Fig. 5. There we take a closer look at the degrees of freedom for the individual parameters, shown as boxplots from all 231805 simulated scenarios in Fig. 5, the capability for determination of the individual parameters can be identified. A value larger than 0.5 (above the dashed grey line) means that this parameter can be determined well (Hou et al., 2018). The spectral albedo values, the surface temperature and the AOD can be determined quite well. When analysing the aerosol components, it is clearly visible that larger particles (e.g. sea salt and dust from bin 003) can be determined reliably in more cases. In the case of carbon compounds, the quantity of hydrophilic aerosols can be determined more frequently than

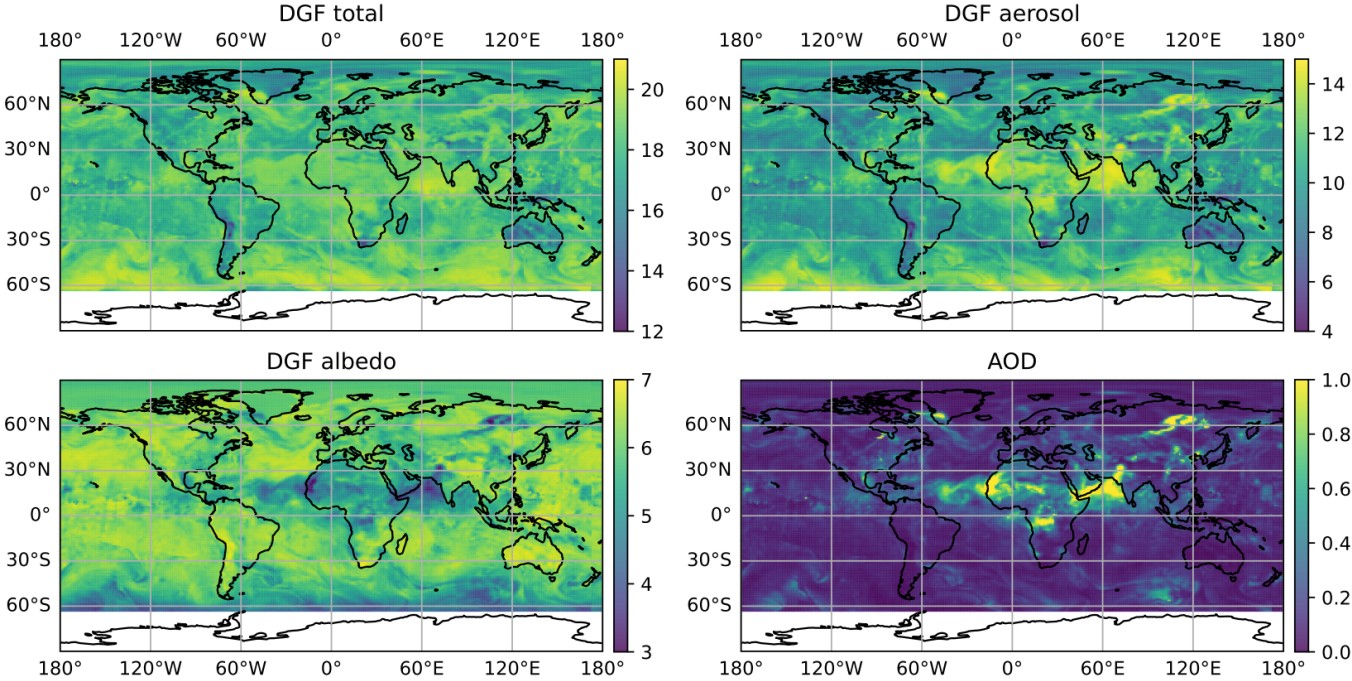

**Figure 4.** Maps showing the distribution of total DGF (top left), DGF for aerosol components (top right), DGF for albedo (down left) and the AOD (down right) on 15.07.2018. Consistent with Fig.3 higher DGF for determination of aerosol components is higher where there is high AOD whereas it is the other way round for the determination of the albedo values. Patterns that primarily reflect air movements over oceans can also be observed through the changes in the degrees of freedom for the determination of aerosol components and albedo values.

hydrophobic aerosols.

These findings are further supported by Fig. 6 where the different DGFs per aerosol component are plotted against the total

AOD. When examining the degrees of freedom of dust and sea salt particles, an increase is observed with rising AOD, asymptotically approaching the value 1. With increasing particle size, a higher initial value at low AOD and a more rapid increase can be observed. The fine mode aerosol components also show an increase in the degrees of freedom with increasing AOD. However, black carbon aerosols show a less pronounced increase than the organic carbon aerosols and sulphates, with black carbon also exhibiting a higher DGF value of 0.45 (hydrophobic) and 0.55 (hydrophilic) at low AOD compared to organic

carbon and sulphate with a value of approximate 0.2. Also, the black carbon saturation levels remain well below 1.

The observations of Fig. 5 and Fig. 6 thus show a higher retrieval capability with increasing AOD and with increasing particle size as well as with decreasing absorption in the fine mode.

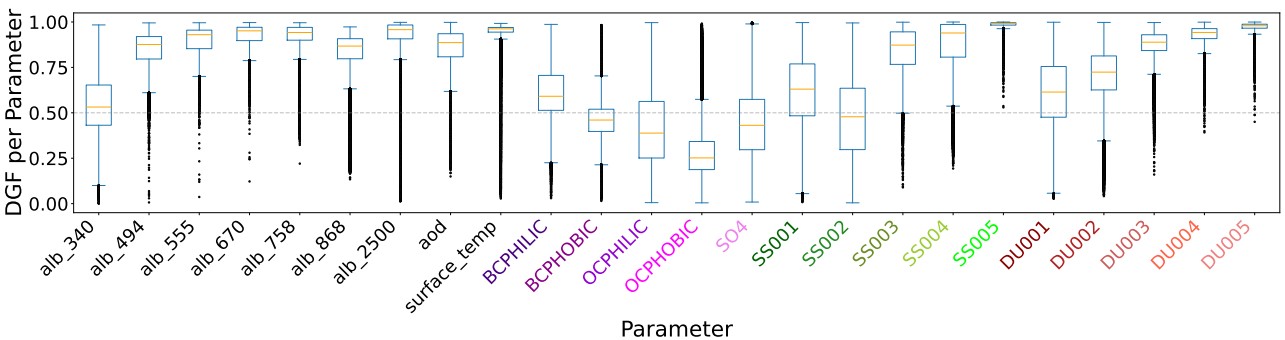

**Figure 5.** Boxplot showing the distribution of DGF values per parameter within all simulated scenarios. A value above 0.5 (dashed grey line) means that this parameter can be retrieved well. The orange line shows the median, the blue box contains the values between the lower and upper quartile, the whiskers indicate the minimum and maximum values and if the whiskers are longer than 1.5 times the box, all values exceeding this range are labelled as outliers in black. The albedo values except the one at $340\mathrm{nm}$ can be retrieved well in most of the scenarios as well as the surface temperature and the AOD. For the aerosol components (fine particle components in purple, sea salt in green and dust in brown), this differs depending on the size and absorption capacity of the components. Larger particles can be determined more reliably than smaller particles, which is clearly visible with dust and sea salt, where the different bins are in ascending order of size. In the case of black and organic carbon, it can be seen that absorbent components can be determined with better quality than (non-)absorbing components.

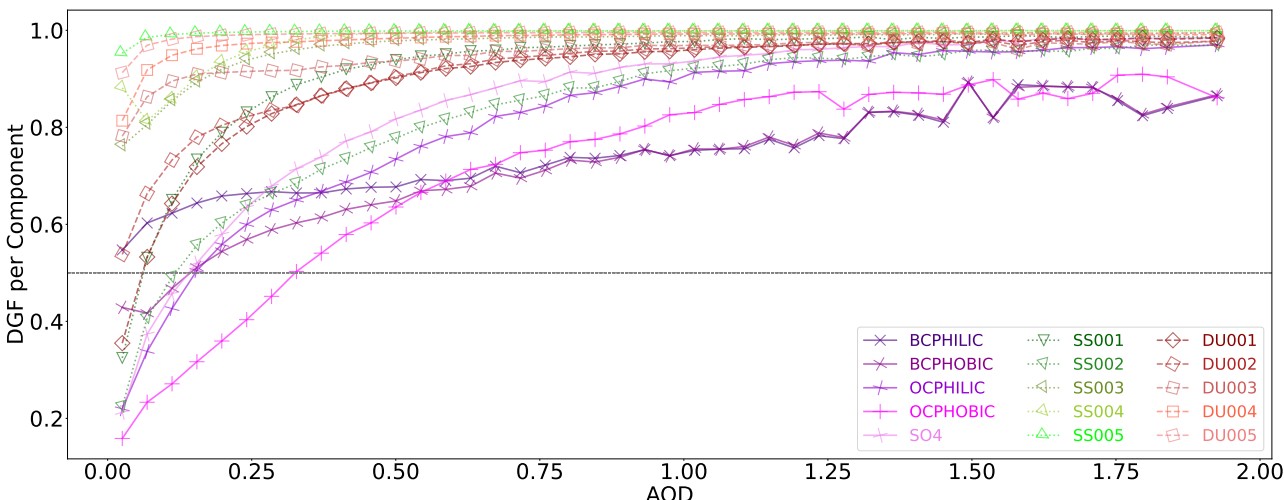

**Figure 6.** Mean values of the binned data of the degrees of freedom per aerosol component plotted against total AOD. The underlying 2D histograms with bin means and standard deviations are shown in A1. For dust (brown, dashed line and squares) and sea salt (green, dotted line and triangles) with increasing AOD an increase of the DGF per component can be seen asymptotically approaching the value 1. With increasing particle size, the initial values at low AOD become higher, and the rise in AOD occurs more rapidly. The different purple graphs with crosses marking the fine particle components carbon components and sulfate exhibit a less steep increase and overall lower DGF values.

## 6 Discussion

This information content study works with simulated radiative transfer calculations matching the radiances arriving at the three instruments in space. For the simulations several simplifications are made to reduce calculation efforts. These may have led to increased quantitative values of the DGF as compared to a real retrieval but should not affect the qualitative statements.

First, the DGF analysis incorporates realistic measurement noise values (the diagonal elements of the measurement error covariance matrices) for all instruments, but it does not include error correlations between the channels (the off-diagonal elements

of the measurement error covariance matrices). This is due to the fact that a comprehensive study of those correlations would require effort which goes beyond this study. Obviously, this simplification holds significant potential to overestimate the information content.

Secondly, the assumption of Lambertian surface reflectance is made which removes one significant source of uncertainty for an aerosol retrieval. However, for the rather coarse spatial resolution of the collocated measurements at 40km x 80km and the

relatively small observation angles effects of the Bidirectional Reflectance Distribution Function (BRDF) should average out in many situations leading to a more isotropic and Lambert like appearance (Zhou et al., 2010) and thus reducing the effect of this simplification.

In this study the measured pre-flight instrument response functions are used for the radiometer SLSTR (channel band widths of $10 - 60$nm in the VIS and $380 - 1000$nm in the TIR range) to model the instrument measurements, while for the narrower

measurements from the spectrometers GOME-2 (<0.5nm resolution) and IASI ($0.5\mathrm{cm}^{-1}$ resolution) we use the simplification of assuming a delta measurement, a measurement at a sharp wavelength, at the center wavelength of the spectral bin instead of the instrumental slit function. This simplification reduces the possible measurement errors and could thus lead to higher DGF in this study compared to a real retrieval, but the effect is expected to remain small.

Additionally, a real retrieval will face several practical issues which violate the assumption of combining observations of the

exactly identical atmospheric volume made at exactly the same time which may reduce DGF values. These include varying time shifts between the satellite overpasses (within the 60 min window allowed), different viewing angle constellations, imperfect co-location of the instruments and calibration inconsistencies / biases between the instruments.

For the vertical distribution of the aerosols we use monthly means for each component on a $1°$x$1°$ grid from MERRA-2. As the vertical distribution changes over time this assumption is not exactly valid in a real case and thus will lead to additional

uncertainty for the retrieval of aerosol components.

Another important aspect is cloud detection. For this simulation study, we assume cloud-free conditions; in a real retrieval, cloud algorithms will be used to detect and mask out clouds. Undetected clouds add further errors and uncertainties and influence the retrieved parameters decreasing the DGF. On the other hand, the three combined instruments provide significant spectral, spatial and angular information, so that strict cloud masking should be possible.

All these simplifications mean limitations for the quantitative results of this study. However, all of these limitations are the same in both the study of the information content of individual instruments and of their combination. Therefore, the general conclusions remain unaffected. This regards the information gain through the combination of the three instruments, as well as

the qualitative behavior of the DGF as a function of the AOD and the capabilities which aerosol components can be determined better or worse. In summary, it can be stated, that although this information content study has some limitations, it clearly shows the potential of the combination of the three instruments IASI, GOME-2 and SLSTR for a retrieval based on the combination. In addition the information content analysis can be useful as a tool to identify optimal sensor combinations and the choice of channels carrying the largest contribution to the information content for the specific target result (e.g. AOD, aerosol composition, surface properties). This information can be extracted form the Jacobian matrix $K$, which contains the sensitivities of each measurement to each variable in the state vector.

## 7 Conclusions

In this paper we have analysed the information content for an aerosol retrieval using a combination of the three instruments SLSTR on Sentinel 3A and B, GOME-2 and IASI both on MetOp A/B/C and the information gain coming from this combination as compared to the individual sensors. The information gained by combining the three instruments for aerosol retrieval was clearly shown and is important for further climate studies.

From the simulation study, it can be concluded that in addition to at best 7 spectral surface albedo values, the surface temperature and total AOD, the sensor combination offers the possibility of determining the mass mixing ratios for 6 to 15 aerosol components and thus also their contribution to the AOD. The number of parameters that can be determined depends both on the AOD (more parameters can be determined with a higher AOD) and on the surface albedo . How well the aerosol components can be determined also depends on the particle size. This means that the mass mixing ratios of aerosol components can be determined more easily with larger particle sizes.

The results presented here show the capabilities to determine spectral ground albedo as well as AOD and the aerosol composition by combining the data from of the three instruments; with this simplified simulation-based study up to 15 DGF for aerosol components can be shown. This is a significant gain in information compared to single-sensor aerosol retrievals. Those currently used single-sensor aerosol retrievals can usually determine total AOD and fine mode AOD, or dust AOD alone, and ground albedo at one wavelength; best instruments (multi-angle polarimeters such as POLDER) allow to invert AOD, fine mode AOD and single scattering albedo (Holzer-Popp et al., 2013). Retrieving more information on aerosol composition opens up further scientific analysis of aerosol-related geophysical phenomena, for example, the transport of desert dust or of aerosols caused by forest fires, due to their occurrence outside their source regions, but also from industrial combustion and the impact of Covid-19 related restrictions on emissions from industry and the transport sector. A more detailed separation of aerosol components will also enable further research into direct aerosol effects on the climate system and into aerosol-cloud interactions and associated indirect effects on regional radiative forcing.

*Data availability.* The used MERRA-2 datasets can be accessed at Global Modeling And Assimilation Office and Pawson (2015a) and Global Modeling And Assimilation Office and Pawson (2015b) and the GOME-2 surface LER database can be accessed at Tilstra et al.
(2017, 2021).

## Appendix A: DGF aerosol components

**Table A1.** Aerosol components used for this study from MERRA-2 (Randles et al., 2017).

| Aerosol Component | abbreviation | dry effective radius in $\mu$m |
|---|---|---|
| Dust Bin 1 | DU001 | 0.64 |
| Dust Bin 2 | DU002 | 1.34 |
| Dust Bin 3 | DU003 | 2.32 |
| Dust Bin 4 | DU004 | 4.20 |
| Dust Bin 5 | DU005 | 7.75 |
| Sea Salt Bin 1 | SS001 | 0.08 |
| Sea Salt Bin 2 | SS002 | 0.27 |
| Sea Salt Bin 3 | SS003 | 1.05 |
| Sea Salt Bin 4 | SS004 | 2.50 |
| Sea Salt Bin 5 | SS004 | 7.48 |
| Hydrophobic Black Carbon | BCPHOBIC | 0.04 |
| Hydrophilic Black Carbon | BCPHILIC | 0.04 |
| Hydrophobic Organic Carbon | OCPHOBIC | 0.09 |
| Hydrophilic Organic Carbon | OCPHILIC | 0.09 |
| Sulfate | SO4 | 0.16 |

*Author contributions.* US carried out the simulations and the analysis of the data and prepared the manuscript with valuable input from TP, MV and HB. US and TP developed the retrieval set-up with the satellite combination. HB has introduced the idea of an information content analysis. MV developed the idea of using MERRA-2 data to get realistic aerosol compositions in the simulations and helped US to integrate
them. US develop the code for the study with support of MV. All authors conceived the research and significantly contributed to the scientific discussions.

*Competing interests.* The contact author has declared that none of the authors has any competing interests.

*Acknowledgements.* The authors gratefully acknowledge the computational and data resources provided through the joint high-performance data analytics (HPDA) project "terrabyte" of the German Aerospace Center (DLR) and the Leibniz Supercomputing Center (LRZ).

We acknowledge the free use of the GOME-2 surface LER database provided through the AC SAF of EUMETSAT. The GOME-2 surface LER database was created by the Royal Netherlands Meteorological Institute (KNMI).

We acknowledge the free use of the Monthly Climatology Emissivity Spectral Data with spatial grids of 0.25-deg lat.-long. based on IASI data of MetOp-A Data from Jun 1, 2007 to Dec 31, 2016, MetOp-B Data from Aug 1, 2013 to Dec 31, 2023 and MetOp-C Data from Jul 1, 2019 to Dec 31, 2023 from Dr. Daniel K. Zhou of NASA Langley Research Center.

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

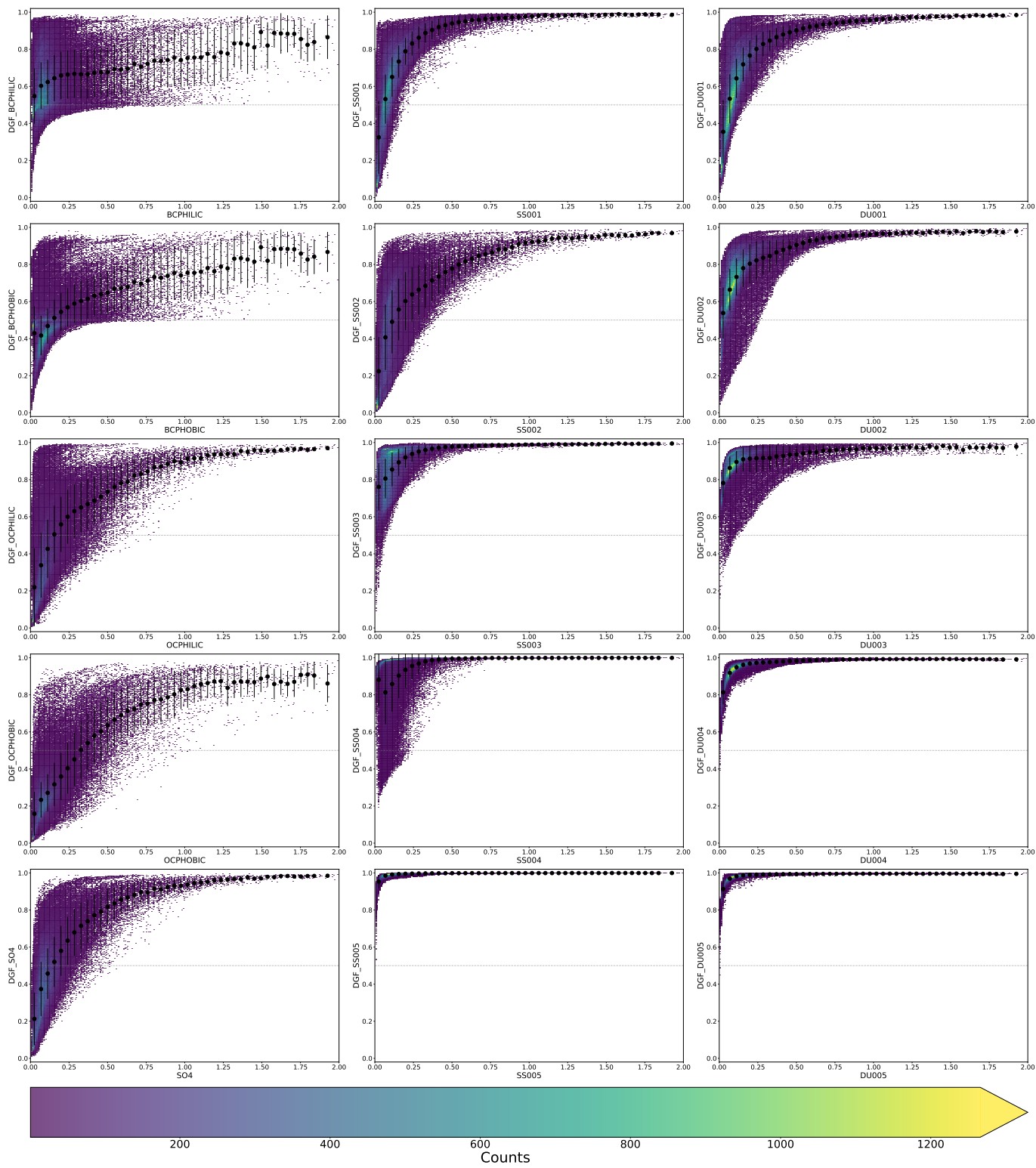

**Figure A1.** 2D-Histgramms showing the DGF for aerosol components in dependency of the 15 different aerosol components. In addition the binned data is plotted in black (dots for the mean value per bin and error-bar showing the standard deviation) .