# Peer review of "Aerosol Composition Retrieval from a combination of three different space-borne Instruments: Information content analysis"

_EGUsphere, 2024_

## Author Response (AR1)

**Author´s response:**

**Reply to Review #2:**

We would like to express our sincere gratitude to the reviewer for their thorough and thoughtful evaluation of our manuscript. We greatly appreciate the positive feedback and the encouraging comments on the overall quality of the study. The insightful reviewer observations have reinforced the importance of our findings, and we are thrilled that the reviewer found our work to be both scientifically sound and well-executed. We have carefully addressed all comments and suggestions in the revised manuscript, which we believe have further strengthened the clarity and quality of the work. The answers to the minor comments (black) can be found below (green) as well as the parts added to the manuscript (blue) as proposed by Reviewer #2. In addition all changes of the initial manuscript based on comments of both Anonymous Referees are tracked in the supplemental material.

**Review of the manuscript titled "Satellite Aerosol Composition Retrieval from a combination of three different Instruments: Information content analysis" by Stöffelmair et. al.,**

This manuscript focuses on retrieval of AOD and its components from the valuable apace-borne instruments such as SLSTR on Sentinel 3A/3B, IASI and GOME-2 on MetOp A/B/C. Further, this study has used SCIATRAN and uses MERRA-2 reanalysis data to study information content of the conducted retrieval and suggested that 6 to 15 degrees of freedom for the determination of aerosol components dependent on AOD and the underlying surface.

This manuscript is well written and all the analysis are discussed in a very appropriate way. The information content analysis presented in this study will be meaningfully supporting the future AOD retrievals over various regions of the globe. I would recommend this manuscript for publication after these minor comments are addressed.

Minor comments:

1. Change the title, as in this study the satellite is being used to detect the AOD and its components, thus change the title as follows "Aerosol Composition Retrieval from a combination of three different space-borne Instruments: Information content analysis". This title will be more appropriate. Done
2. Please correct the line 4 as "A simulation study has been carried out to analyse the information…" Done
3. The introduction is very well written and easy to follow.
4. The theory of information content and Optimal Estimation is also discussed in a very understandable way.
5. Please add a paragraph focusing only why SCIATRAN in particular is considered in this study and what are the advantages section 3.1 "Radiative transfer forward model". We added the following paragraph in this section:

   SCIATRAN is a well test radiative transfer model that can be used to compute radiances over a broad spectral range, in particular the UV-VIS and IR ranges needed for this study. A second reason is SCIATRAN's option to take into account different aerosol components as defined in MERRA-2.

6. Similarly, please add a paragraph focusing only why only these satellite instruments are used in this study and advantages of selecting these instruments in section 3.2 "Satellite Measurements and Observation Vector". We added to the first paragraph, more details to

 In addition the spatial and temporal overlap of their measurements plays an import role in that choice. Moreover, the chosen combination of instruments allows for the possibility of a long time series through their predecessor and successor instruments.

7. Please improve the titles of all the figures, so that the readers can easily follow the figures. Captions are now more detailed, they can be seen in the attached track changes file.

8. Discuss why there is no data information available over Antarctica, may be add two to three sentences in the result section. Is added in the results section:

   Data south of 62°S were measured at solar zenith angles of over 90° on this date in north-hemispheric summer. Under these conditions, the amount of reflected UV and VIS radiation is too low and the retrieval is not performed.

9. Separate the "Discussion and Conclusion" sections into two sections and also in discussion section add a paragraph about how the information content analysis can be useful for the existing and future satellite missions. Discussion and conclusion are separated and the following paragraph is added in the discussion section:

   In addition the information content analysis can be useful as a tool to identify optimal sensor combinations and the choice of channels carrying the largest contribution to the information content for the specific target result (e.g. AOD, aerosol composition, surface properties). This information can be extracted from the Jacobian matrix K, which contains the sensitivities of each measurement to each variable in the state vector.

At the end, I would like to mention that this manuscript is short, concise, and the results are well discussed and contains valuable information for the enhancement of current and future retrieval process of AOD and associated components. I wish the authors in advance Merry Christmas and successful start to the new year!

**Reply to Review #3:**

We would like to sincerely thank the Reviewer #3 for their detailed and constructive feedback. We greatly appreciate the time and effort that went into the review and the thoughtful, critical insights provided. Although some of the comments highlighted areas for improvement, we found them to be helpful in refining our study. The suggestions have allowed us to enhance the clarity of our methodology and expand on important aspects of the discussion. We believe that the revisions made in response to the reviewer#3 comments have significantly improved the manuscript. The answers to the comments (black) can be found below (green) as well as the parts added to the manuscript (blue) as proposed by Reviewer #3. In addition all changes of the initial manuscript based on comments of both Anonymous Referees are tracked in the supplemental material.

**Review of "Satellite Aerosol Composition Retrieval from a combination of three different Instruments: Information content analysis", by Stoeffelmair et al.**

This paper presents an information content analysis of simulated top-of-atmosphere measurements spanning the spectral range from the UV through to the thermal infrared for the determination of atmospheric aerosol properties (and surface albedo). The simulated measurements are based on the combination of the GOME-2, SLSTR and IASI instruments and the authors state that their analysis is a step towards the development of a retrieval utilising this combination of instruments to retrieve

information on aerosol composition, which is an important in determining the direct and indirect radiative impacts of aerosol.

Overall, the analysis performed is sound and worthy of publication, however the simplistic representation of the instruments involved, representation of measurement and forward model uncertainty and surface reflectance limit the applicability of the results in predicting the performance and information content of an actual retrieval system applied to real measurements. Thus, I feel that the primary conclusion the authors draw from the work - that this combination of instruments can provide considerable constraint on aerosol composition (in addition to AOD and surface albedo), needs tempering somewhat with the following considerations:

1. The assumption of Lambertian surface reflectance removes one of the key sources of uncertainty when using visible and near-IR measurements for aerosol retrieval. Simulations made using this assumption will thus overestimate the information content of SLSTR measurements available for constraining aerosol properties.
2. The use of simple instrument noise estimates to set the covariance matrices used in the analysis ignores forward model error and correlations between the elements of the measurement vector, both of which will decrease the information available to constrain aerosol properties.
3. The authors also seem to neglect the vertical distribution of aerosol, which can have an impact on TOA radiance significantly exceeding aerosol composition in both the UV and TIR. Dealing with this in a real retrieval scheme will again reduce the information available for constraining aerosol composition.
4. The simplistic way that the measurements from the individual instruments is modelled (neglecting the problems of co-location and matching the very different viewing geometries and spatial sampling of the instruments cited, as well as not attempting to accurately model the spectral response of the real instruments) also neglects important sources of error which would be present in a real retrieval scheme and further abstracts the simulations performed from what might be expected from real measurements.
5. The authors never address another significant source of error and complexity in any aerosol retrieval scheme, which will be even more difficult to deal with when combining three instruments with completely different sensitivities and sampling - clouds!

To be clear, none of these points negate the value of the analysis presented and the conclusion that a combination of measurements like those provided by GOME-2, SLSTR and IASI do provide information on aerosol composition, which is largely lacking in the current generation of satellite-based aerosol products, and which is of great importance in better constraining the role of aerosol in climate. However, in its current form, I feel the paper somewhat over sells the potential of the proposed retrieval approach and I believe the authors need to be more up-front about the limitations of their analysis. There is no way a scheme using this combination of instruments will provide up to 15 independent pieces of information on aerosol composition (and, indeed, current schemes utilising the individual instruments do not match the performance suggested by Figures 1 and 2).

For this reason, I believe this paper should be published in AMT, provided the text is modified to make the limitations of the analysis more clear and to more explicitly state the qualitative nature of the results and conclusions (as opposed to a quantitative analysis of the information content of a

scheme combining these three instruments). It would also be interesting and a lot more informative if the authors spend some time to investigate the impact of their assumptions and simplifications on the information content. What is the dependence of the degrees of freedom on the assumed measurement covariance and the relative weighting of each instrument, for instance, or what impact do differences in the area sampled by each instrument have?

Thank you for pointing to the limits of this study, we have added a separate discussion section:

[revised manuscript text omitted]

In addition to this general recommendation I also have the follow specific corrections and points on the text:

*Pg1, ln4:* Inconsistent use of tense - a simulation study "has been", or "is" carried out (rather then "will be"). This is changed.

*Pg1, ln14:* A *brief* definition of direct, semi-direct and indirect effects should be provided on their first use. We agree, and added a brief definition:

Their direct (influencing the radiation budget directly by scattering, absorbing or emitting radiation), semi-direct (effects on cloud properties by heating or cooling the atmosphere) and indirect effects (affecting cloud properties through acting as a condensation nuclei or ice nucleating particles) depend not only on the aerosol abundance and geospatial distribution but also on the aerosol chemical composition (Boucher et al., 2013; Kaufman et al., 2002).

*Pg1, ln17:* There is a unnecessary ellipsis (...) here. Corrected

*Pg1, ln21:* "also depends" rather than "depends also". Done

*Pg1, ln21:* Final clause of this sentence, beginning "because different aerosol...", is superfluous and can be deleted. Done

*Pg2, ln28:* Combine sentence beginning "Observational data" with the previous one and reword to ", so observational data are important for validation and assimilation purposes." Done

*Pg2, ln29:* Replace the two sentences starting "It is not sufficient..." with "It is not sufficient to constrain just the quantity and distribution of aerosol, composition information is also needed if we want to reduce uncertainties on climate forcing due to aerosol. Hence, there is an important climate research need for global monitoring of aerosol composition from satellite measurements." Done

*Pg2, ln35:* "there is not enough" (rather than "there are not enough"). Corrected

*Pg2, ln36:* Insert "properties" after the word "surface". Done

*Pg2, ln39:* Insert "and" between "AOD" and "composition". Done

*Pg2, ln40:* "will be developed" rather than "shall be applied". Done

*Pg2, ln44:* The term "aerosol component" needs to be defined on first use (at least within the body of the text). Definition added:

Atmospheric aerosol is typically described by a mixture of a manageably small number of representative components. A component groups particles with similar characteristics (chemical composition, size range, shape and corresponding optical properties). Further we differ between organic and black carbon, sulfates and sea salt and mineral dust at different size bins (Kinne et al., 2006; Randles et al., 2017).

*Pg2, ln44:* The paragraph on this line does not scan well and needs to be rewritten. If the authors are trying to justify the need for their proposed algorithm (which, I guess, is essentially an extension of the idea explored with SYNAER by extending to include the thermal-IR), when the GRASP algorithm exists, this can surely be done more succinctly and clearly.

Actually, rather than GRASP, the authors should be more concerned with explaining how their proposed retrieval scheme will differ from/improve upon the PMAP (Polar Mulit-sensor Aerosol Optical Properties) algorithm, which is operationally run by EUMETSAT and combines measurements from GOME-2, AVHRR and IASI. This algorithm needs to at least be referenced (perhaps PMAP is a direct development of the work of Hasekamp and Landgraf (2005), which the authors reference later in this section, but I'm not sure!) The paragraph is rewritten and PMAp (which was mentioned later) is added here, it is also moved after the part where we describe the different aerosol algorithms:

SYNAER works with predefined aerosol mixtures (fixed mixes of 4 different aerosol components), determines the best fitting mixture and not individual aerosol components (Holzer-Popp et al., 2008). PMAp works with aerosol classes like oceanic, industrial, biomass and dust with different refractive indices and different size distributions (Grzegorski et al., 2021). Another aerosol component algorithm is GRASP/Component (Li et al., 2019, 2020; Zhang et al., 2021; Dubovik et al.,2021a), which is based on the multi-axis and polarimetric data from POLDER/PARASOL.

*Pg2, ln57:* "additional channels in the visible range" - additional to what? Corrected:

"additional channels in the visible range and" is deleted

*Pg2, ln58:* Reword sentence to "IASI is mostly sensitive to mineral dust and larger particles". Also, a more general point to note is that IASI and GOME-2 are also sensitive to elevated stratospheric sulphate aerosol loadings, when compared to SLSTR. We agree, changed and added: In addition IASI and GOME-2 are also sensitive to elevated stratospheric sulphate aerosol loadings.

*Pg3, ln68:* Replace sentence starting "This is made possible..." with "The algorithm proposed here has the potential to be applied to predecessor instruments: (A)ATSR(-2) for SLSTR, GOME and SCIAMACHY for GOME-2 and HIRS for IASI, which provide temporal..." Done

Also, I would imagine imminently upcoming instruments also lend themselves to this retrieval approach. Sentinel-4 perhaps? We added the planned successors to the used instruments to the text: "As all the instruments have planned successors the time series can be continued at least until 2035. The planned successor to GOME-2 is UV/VIS/NIR/SWIR Sounding (COPERNICUS Sentinel-5 UVNS) and to IASI it is the Infrared Atmospheric Sounding Interferometer – New Generation both on MetOp second Generation (Holmlund et al., 2017) and SLSTR will be continued on Sentinel 3C and 3D (World Meteorological Organization (WMO), a, b)." Sentinel 4 would be difficult because there the combination with IR and also global coverage in missing.

*Pg3, ln76:* The paragraph starting here would more sensibly be placed earlier in this section. We moved the paragraph originally starting in ln44 behind this one, to leave the explanation of the used

instruments before this paragraph so it is clear why those and not all possible instruments for aerosol retrieval are described.

*Pg3, ln87:* Here the topic of information content and degrees of freedom is launched into without any explanation of what is meant by these phrases. The mathematical definitions can wait until later in the paper, but a simple explanation of their meaning is needed here. Added in ln109: An information content study shows the amount and type of information which can be extracted from the data. In this context, the degrees of freedom (DGF) represents the number of parameters that can be retrieved.

*Pg4, ln113:* Replace "the reflectance" with "spectral reflectance and brightness temperatures" (or whatever is appropriate, but you are surely not using top-of-atmosphere reflectance as a measurement in the thermal-IR). Done

*Pg4, ln116:* Both "cost function" and "minimised" need to be defined. We have deleted this sentence, following your next comment to only leave key word which are necessary in this study.

*Pg4, ln120 / equation (2):* There is a lot to unpick here. Firstly, I'm not sure why you're introducing an iterative update to a state vector, since the paper is not describing a retrieval or optimisation scheme. That not withstanding, you also do not define S_eplison or where your initial guess at x_i might come from. You might also consider making it clear that superscript "T" and "-1" refer to matrix transpose and inversion respectively. We agree and so we have deleted Eq 2 and Eq 3 and defined $S_{epsilon}$. Where $x_i$ came from is described in section 3.4, in Section 2 we only like to introduce the theory.

*Pg4, ln122:* In practice the Jacobian matrix is made up of the derivatives of the forward model wrt to the state vector, not the measurements. Corrected

*Pg4, ln123:* Define x-hat. Done

*Pg4, ln125 / equation (3):* Note that this equation is the linear approximation of error covariance, and thus will only be valid if evaluated at the true state for a non-linear system. Is deleted because of your comment 3 above this.

*Pg5, ln128 / equation (4):* The averaging kernel "A" is a key concept/quantity for this paper, so it's probably worth naming. Also, further explanation of what it represents would be desirable. For instance, you note that the diagonal elements denote the sensitivity of a retrieved parameter to its true value, but what do off diagonal elements of this matrix represent? The averaging kernel matrix A was named in ln127 already, we give additional details now:

For the information content analysis the averaging kernel matrix

A =…

is used to calculate the Degrees of Freedom (DGF). It represents the partial derivation of the retrieval state vector $\hat{x}$, which is the estimate of the true state vector x obtained by the optimal estimation algorithm, with respect to x. $S_a$ is the error covariance matrix corresponding to a priori state vector $x_a$. The error covariance matrix for the measurements $S_\epsilon$ contains the instrument measurement uncertainties. K is the Jacobian matrix consisting of the partial derivatives of each measurement, in this study each calculated y value from the forward model, with respect to each state element ($K_{ij} = \partial y_j / \partial x_i$ ). The superscripts "T" and "-1" refer to matrix transpose and inversion.

And later:

The diagonal element values of $A_{ii}$ are in the range of 0 (no information on $x_i$) to 1 ($x_i$ can be fully determined) and characterize the sensitivity of each retrieved parameter to its truth. This makes the DGF a good indicator of the number of parameters that can be determined in retrieval. The off-diagonal elements describing the cross-correlation between the parameters indicate how strongly the estimate of one parameter depends on other parameters.

*Pg5, ln141:* Remove "for radiative transfer and". Done

*Pg5, ln142:* Replace "observations from UV to TIR" with "observations across the UV to TIR". Done

*Pg5, ln149:* "the MERRA-2 model comprises precomputed" (plural). Done

*Pg6, ln172:* I'm not sure what you are referring to by "infrared camera". If you mean the sensor which converts the incoming thermal radiation to an electric current, I think that is an implicit component of a Michelson interferometer. IASI contains an additional infrared camera for other measurements, but it was not used. Consequently this part of the text is deleted to avoid confusion.

*Pg6/7 - Section 3.2:* Several details aren't clear from this section and should be explained:

- Are radiative-transfer calculations performed at the native resolution of each simulated instrument and then averaged onto the GOME-2 grid, or are all calculations performed on the GOME-2 grid from the start? (I assume the latter).
  The radiative-transfer code is one-dimensional, assuming horizontally homogeneous conditions and not considering instrument pixel size. Only the measurement data will be averaged on the common grid for the planned retrieval. For clarification this sentence is added to the paragraph: The DGF analysis is made with a perspective of a synergistic retrieval algorithm for those three instruments using atmospheric radiative transfer simulations, which do not include an instrument model, i.e. they are done monochromatically at central spectral bins and one-dimensional.
- For IASI, are you essentially simulating L1C data by performing the radiative transfer calculations assuming a flat instrument response function? In this study we used the assumption of monochromatic measurements per bin as described under the last comment.
- In the case of GOME-2, are you simulating realistic GOME-2 spectra, or more simplified "GOME-2 like" spectra. Saying you use "a wavelength step of approximately 10 nm" is quite vague. We use a simplified "GOME-2" like spectra using around every tenth spectral bin of the GOME-2 spectra and assuming delta spectral shape measurements.
- I feel the description of the measurement errors/uncertainties should be included in the description of the observation vector. Is added

*Pg6, ln173:* A Michelson interferometer measures an interferogram (hence the name), which is converted to a spectrum through a Fourier transform operation, not the other way around. Corrected

*Pg7, Section 3.3:* I am surprised that you don't say how many elements there are in your state vector in total. This is key parameter, as it defines the maximum DGF value for your model. Also, if I understand correctly, the only variation in aerosol height profile in you model is through differences in the profile of each component in the MERRA-2 database? In this case, you're sensitivity to aerosol component is actually a mixture of the composition/optical properties of each component and it's height profile. Please explain. The reviewer is right, we added the number of parameters (25) and an explanation is added about the sensitivity to the height profiles:

This means that the sensitivity to aerosol components is a mixture of the composition/optical properties of each component and its typical height profile.

*Pg7, Section 3.4:* Values of the a priori state vector are not relevant to your analysis (x_a doesn't appear in equation 4). What is relevant is the a priori covariance matrix, but you don't mention this anywhere. Please correct. Corrected. The section title is changed to: "Apriori values and error covariance matrix used in Optimal Estimation" and this part is added:

The apriori error covariance matrix Sa has the following diagonal elements: 0.2 for the constrains for the albedo values, 5.0 for the AOD, 30 (K) for the surface temperature and 1 for the scaling factors for the mass mixing ratios of the aerosol components. We use Sa as a diagonal matrix. Consequently, all off-diagonal elements are set to zero, because the constrain of one parameter to another one is not known.

*Pg7, ln206:* Replace "aerosol retrieval from the combination of three instruments" with "the forward model arrangement described above". Done

*Pg7, paragraph beginning on ln212:* I don't understand the procedure described here. Firstly, why regrid the MERRA-2 mixing ratios from their native resolution to to 1x1 degree lat-lon grid?

Also, what is the purpose of calculating monthly means of mass-mixing ratios and then normalising them? Please make it clear what data MERRA-2 actually provides you and what you are converting this into using these calculations. We reformulated the paragraph:

To account for a representative range of the true state parameters, global scenarios derived from the MERRA-2 reanalysis are utilized. As we like to do this analysis on a 1°x1° grid, the mass mixing ratios from MERRA-2 are re-gridded from 0.5°x0.625° to 1°x1° using bilinear interpolation. The satellite overpasses are at 9:30 am for the MetOp satellite, with GOME-2 and IASI on it, and 10:00 am for Sentinel 3A and 3B, with SLSTR on board. Consequently we select for each time zone, every 3 hours in the MERRA-2 data, the nearest to 9:30 am local solar time. The scaling factors are then calculated by normalizing the profiles to $1 kg kg^{-1}$, as described in Section 3.3. The scaling factors are used for the simulation study to calculate the simulated data. The relative humidity is taken from MERRA-2 and is not retrieved in this study.

*Pg8, paragraph beginning on ln220:* So, if I understand correctly, you are ignoring the range of viewing angles observed by the different instruments, and the temporal difference between the Sentinel-3 and Metop platforms? This is a substantial simplification of the actual measurement system and should be noted as such. Is added and explained:

We use solar angles calculated with the python package pvlib (Anderson et al., 2023) at a local solar time of 9:30 am which corresponds approximately to the satellite overpasses. The minimal solar movements between the satellite overpasses of MetOp and Sentinel-3, which are at most half an hour apart, are neglected in this context. For the satellite viewing geometry, we use the simplified case that all instruments measure as close as possible near nadir above the point under consideration. That means 0° viewing zenith angle for GOME-2 and IASI and 6° for the nadir view of SLSTR. For the surface albedo the GOME-2 surface LER climatology data (Tilstra et al., 2017, 2021) is used as a priori information.

*Pg9, paragraph beginning on ln 224:* This description of measurement uncertainties belongs in section 3.2. Here would be the place to include forward model error description, if you'd included any. We agree, and moved the measurement uncertainty description  into the different subsection of

3.2. As SCIATRAN shows good agreements with other radiative transfer models and with space and ground-borne measurements (Rozanov, 2014, doi: 10.1016/j.jqsrt.2013.07.004), the forward model error and the uncertainties of other assumed properties (eg. gas absorption cross section) are negligible compared to the measurement errors and the added uncertainties through the retrieval process (including aerosol optical properties), so we do not include a forward model error.

*Pg9, ln232:* What is described here is not interpolation or regridding. You are simply sub-sampling the data, with one 1x1 degree box extracted from each 10x10 degree region. Corrected to:

For this analysis we select a subset of 1°x1° grid boxes to get a 10°x10° grid in order to consider the best possible coverage of different aerosol compositions with a reduced amount of data.

*Pg9, ln245:* Am I correct in think the DGF for "aerosol components" is simply the total of the diagonal elements of the averaging kernel corresponding to the MERRA-2 scaling factors for each component? Please be more explicit. Corrected:

The degrees of freedom for all aerosol components is defined as the sum of the diagonal elements of the averaging kernel matrix corresponding to the MERRA-2 scaling factors for each component; it provides the result targeted at the primary goal of this study, determining the aerosol composition.

*Pg11, Figure 5:* Some visual categorisation of the aerosol components into broad types (like dust, sea-salt) would be helpful here - maybe through colouring the labels? We use now the same colors like in figure 6 for the labels, using the same color family for a broad aerosol type and describing this in the caption.

[Figure]

**Figure 5.** Boxplot showing the distribution of DGF values per parameter within all simulated scenarios. A value above 0.5 (dashed grey line) means that this parameter can be retrieved well. The orange line shows the median, the blue box contains the values between the lower and upper quartile, the whiskers indicate the minimum and maximum values and if the whiskers are longer than 1.5 times the box, all values exceeding this range are labelled as outliers in black. The albedo values except the one at 340nm can be retrieved well in most of the scenarios as well as the surface temperature and the AOD. For the aerosol components (fine particle components in purple, sea salt in green and dust in brown), this differs depending on the size and absorption capacity of the components. Larger particles can be determined more reliably than smaller particles, which is clearly visible with dust and sea salt, where the different bins are in ascending order of size. In the case of black and organic carbon, for example, it can be seen that absorbent components can be determined with better quality than non-absorbent components.

*Pg12, Figure 6:* Similar comment to Figure 5. We try to do this here through the taking same color families (purple, orange and green) and the same markers, but rotated. For more clarity we adapted also the caption.

[Figure]

**Figure 6.** Mean values of the binned data of the degrees of freedom per aerosol component plotted against total AOD. The underlying 2D histograms with bin means and standard deviations are shown in A1. For dust (brown, dashed line and squares) and sea salt (green, dotted line and triangles) with increasing AOD an increase of the DGF per component can be seen asymptotically approaching the value 1. With increasing particle size, the initial values at low AOD become higher, and the rise in AOD occurs more rapidly. The different purple graphs with crosses marking the fine particle components carbon components and sulfate exhibit a less steep increase and overall lower DGF values.

*Pg13, ln300:* Where does this mention of soil-type come from? Do you mean surface albedo? Yes, its corrected now.

*Pg13, ln304:* I don't think you can claim you've used realistic measurement noise. In general, this paragraph does not go far enough in acknowledging the limitations of your analysis, particularly with regard to simulating a retrieval scheme applied to real-world measurements. The limits are now discussed in more detail in a separate discussion section, which you can find in the beginning of this answers to your general comments.

*Pg13, ln307:* Spelling "varying". Done

*Pg13, ln310:* Replace "data of the three instruments" with "data from the three instruments". Done